# Association between Sexual Behavior and Depression in South Korean Adolescents: A Cross-Sectional Study

**DOI:** 10.3390/ijerph18084228

**Published:** 2021-04-16

**Authors:** Hyunkyu Kim, Wonjeong Jeong, Sungin Jang, Youseok Kim, Euncheol Park

**Affiliations:** 1Department of Preventive Medicine, College of Medicine, Yonsei University, Seoul 03722, Korea; healbot@yuhs.ac (H.K.); JANGSI@yuhs.ac (S.J.); 2Institute of Health Services Research, Yonsei University, Seoul 03722, Korea; wjjeong@yuhs.ac (W.J.); KYSMD@yuhs.ac (Y.K.); 3Department of Psychiatry, College of Medicine, Yonsei University, Seoul 03722, Korea; 4Department of Public Health, Graduate School, Yonsei University, Seoul 03722, Korea; 5Department of Hospital Administration, Graduate School of Public Health, Yonsei University, Seoul 03722, Korea

**Keywords:** depression, suicide, sex, sexual behavior, adolescents, Korea Youth Risk Behavior Web-Based Survey

## Abstract

Adolescent depression and suicide have become leading public health and socioeconomic problems. Determining the connection between adolescent behavior and depression can inform strategies to reduce the prevalence of depression and suicide. We investigated the association between sexual behavior and depression in South Korean adolescents. Data for this cross-sectional study were obtained from the 2017–2019 Korea Youth Risk Behavior Web-Based Survey. Data of 178,664 subjects were analyzed using chi-square tests and multivariate logistic regression. After adjusting for covariates, the prevalence of depression was found to be higher in subjects with experience of sexual intercourse (adjusted odds ratio = 1.71, 95% confidence interval = 1.59–1.83 in boys; adjusted odds ratio = 1.47, confidence interval = 1.33–1.61 in girls). On categorizing subjects into two groups based on suicidality, subjects with sexual intercourse experience had higher odds ratios for depression with suicidality (aOR:2.16 in boys, aOR:1.80in girls) than depression without suicidality (aOR:1.49 in boys, aOR:1.25 in girls). We identified the relationship between sexual behavior and the prevalence of depression; adolescents with experience of sexual intercourse were more likely to have depression with suicidality. Further research using prospective designs should serve as the basis for appropriate sex education policies to manage the relationship between sexual behavior and depression.

## 1. Introduction

Globally, depression and suicidal behavior in adolescents are becoming major social problems with huge economic repercussions [1]. In the United States, 15.7% of adolescents reported more than one symptom of major depression in 2019; this is a rapid increase from the 2004 rate of 9% [2]. The Centers for Disease Control and Prevention reported that in 2019, 18.8% of U.S. high school students reported having seriously considered suicide and 8.9% attempted suicide; this rate had increased since 2009 [3]. As demonstrated by these statistics, suicide is a serious concern, one that is now the second leading cause of death of adolescents in the United States [4,5].

The situation in the Republic of Korea is similar [6]. In a 2016 survey of school students, 19.7% of male students and 27.8% of female students reported experiencing depression [7]. According to the South Korean government, the leading cause of death in teenagers in 2019 was suicide, with a prevalence of 5.9 per 100,000 [8]. Thus, the appropriate management of depression and suicide is an important public health concern. Adolescence is a period of physical, mental, and emotional growth when values and personality are malleable; therefore, the identification and management of factors affecting the occurrence of depression in this population is of the utmost significance.

As previously mentioned, adolescence is a critical period for not only physical growth but also psychosexual development, including sexual identity and orientation. Thus, sexual behaviors during this period can have significant mental and social impacts. In the past, studies have examined the physical and psychological effects of adolescent sexual behavior. In this context, Vasilenko et al. suggested a conceptual framework for the effects of sexual behavior on mental, physical, and social health [9]. They posited that sex could be a more negative experience for early adolescents than for older teenagers. Another study on seventh to twelfth graders in the United States reported that sexual behavior increased the odds ratios (ORs) of depression and suicidality, and that this effect was compounded when combined with drug abuse [10]. These studies, though limited in scope and sample sizes, suggest that sexual behavior in adolescents can cause, or at least correlate with, depression.

Sexual behavior can have a greater impact on adolescents than other age groups, and the intensity can vary depending on the cultural and social climate, such as Confucianism [11]. South Korea is relatively conservative with regard to sex; sexual activity among youths is viewed particularly negatively, as demonstrated by a survey comparing the sexual behaviors and consciousness of South Korean and Dutch adolescents [12]. The concept of chastity is important to South Koreans, and there is a general trend toward abstinence from sexual behaviors in adolescents. However, considering the rise in sexual activity among adolescents, as demonstrated by the decrease in the average age when sexual activity is initiated—from 14.3 years in 2011 to 13.3 years in 2014—the impact of adolescent sexual behavior on psychiatric symptoms such as depression is increasing [13]. Thus, it is necessary to conduct a detailed and large-scale examination of the relationship between sexual behavior and depression in adolescents.

The aim of this study was to investigate the association between sexual behavior and depression in a large sample of South Korean adolescents, after adjusting for covariates associated with depression prevalence. Furthermore, we aimed to identify differences in the suicidality of individuals with depression based on the presence or absence of sexual behavior.

## 2. Materials and Methods

### 2.1. Study Population and Data

The data analyzed in this study were obtained from the 2017–2019 Korea Youth Risk Behavior Web-Based Survey (KYRBS), conducted by the Ministry of Education, Ministry of Health and Welfare, and Centers for Disease Control and Prevention. The purpose of this survey, which targets middle and high school students, is to evaluate the health status and health behavior of South Korean adolescents in order to provide basic data for the formulation of health policies. The anonymous survey is conducted annually with about 400 high schools and 400 middle schools. Survey items may change slightly every year. The survey was designed to represent Korean middle- and high-school students; thus, a complex survey design was introduced considering selection probabilities, survey non-responses, and post-stratification. Weight values are suggested for combined analysis for several years.

### 2.2. Measures

#### 2.2.1. Depression

Students were asked whether they had experienced depressive feelings that interfered with daily life for more than two weeks in the previous year. As the KYRBS did not directly measure depression severity, we divided subjects into two groups based on reported suicidality: (a) Those with depression who had not experienced any suicidal ideation, planning, or attempts; and (b) those with depression who had experienced suicidal ideation, planning, or attempts within the previous 12 months.

#### 2.2.2. Sexual Behavior

Sexual behavior was assessed through questions such as whether the subjects had ever engaged in sexual intercourse and whether they had received sex education through any means in the previous 12 months. Subjects who indicated that they had experience of sexual intercourse were asked about contraceptive methods.

#### 2.2.3. Covariates

Age, socioeconomic status, academic grades, and family structure were included in the sociodemographic covariates. Socioeconomic status was categorized as high, middle, or low, and based on family structure, we classified subjects as either having one or both parents or none. The analysis was also adjusted for health-related covariates including smoking status, alcohol use, perceived stress level, and self-reported health status.

### 2.3. Statistical Analysis

Chi-square tests were used to analyze and compare the variables. To examine the relationship between sexual behavior and depression, we conducted a multivariate logistic regression analysis after adjusting for covariates. Subgroup analyses were performed to investigate the combined associations of sexual behavior and other covariates with depression. Subjects with depression were divided into two groups by the presence of suicidality, and dependent subgroup analysis was performed to compare the odds ratios of the two groups. The results are presented as ORs and 95% confidence intervals (CI) to compare the prevalence of depression. The analyses were performed based on complex survey design with stratified sampling variables (strata) and weighted variables suggested by the KYRBS. All analyses were carried out using SAS software version 9.4 (SAS Institute, Cary, NC, USA), and *p*-values < 0.05 were considered statistically significant.

## 3. Results

The general characteristics of the study population stratified by gender are presented in Table 1. A total of 178,664 subjects, including 91,309 boys and 87,355 girls, were included in the analysis. Among them, 20.7% of boys and 31.3% of girls reported having experienced depression in the previous year. Subjects who had sexual intercourse experience reported higher rates of depression, regardless of gender. Older students reported higher rates of depression than their younger counterparts. Low socioeconomic status was associated with a higher prevalence of depression than middle and high socioeconomic status. Higher perceived stress was associated with a greater prevalence of depression. Family structure, school grade, alcohol status, smoking status, physical activity, and self-reported health status were additionally identified as being statistically significantly associated with depression. Experience of sex education was not statistically correlated with depression.

Table 2 shows the results of the multivariate logistic regression analysis between depression and sexual intercourse experience. After adjusting for covariates, the prevalence of depression was higher in subjects of both genders with sexual intercourse experience (adjusted OR: 1.71, 95% CI: 1.59–1.83 in boys, adjusted OR: 1.47, CI: 1.33–1.61 in girls). After adjusting for covariates, higher age was associated with lower depression. Regardless of gender, the absence of both parents was statistically associated with depressive experience. Other covariates including socioeconomic status, school grade, alcohol/smoking status, physical activity, perceived stress level, self-reported health status, and sex education experience were also correlated with depression, as shown in Table 2.

Subgroup analyses were conducted to assess the combined associations of sexual intercourse experience and other sociodemographic variables with depression, as shown in Table 3. Sexual experience was correlated with a high prevalence of depression in every subgroup except sex education. Subjects with sexual intercourse experience showed higher ORs for depression when they attended sex education. However, this was not statistically significant in girls without sex education experience (OR: 1.16, CI: 0.95–1.43).

The results of multivariate logistic regression analyses, conducted after dividing subjects into the depression with/without groups, are presented in Table 4 and Figure 1. Regardless of gender, subjects with sexual intercourse experience showed higher ORs for depression with suicidality than depression without suicidality.

## 4. Discussion

In this investigation of a sample of South Korean adolescents, we identified that depression was associated with experience of sexual intercourse. Furthermore, we found that those with sexual intercourse experience had a stronger association with depression with suicidality than with depression without suicidal ideation, planning, and attempts. These results did not differ by gender and remained consistent after adjusting for covariates and subgrouping by sociodemographic variables.

Previous studies have demonstrated associations between sexual behavior and depression or suicidal ideation in smaller samples. Hallfors et al. reported a statistically significant relationship between individuals with experience of sex and depression (OR: 2.65) and suicidal ideation (OR: 2.53) [10]. Depression was assessed with the Center for Epidemiologic Studies Depression Scale, with the questions pertaining to the past year, similar to our study.

Compared to adults, depression in adolescents has different characteristics, including conduct behaviors [14]. It is widely known that risky behaviors, including sexual activity and drug abuse, are consequences of depression. In fact, Kessler et al. hypothesized that such behaviors act as self-medication in adolescents with depression [15]. Conversely, studies suggest that these behaviors have biological and psychological effects that increase the risk of depression [16]. According to Hallfors et al., it is more reasonable to conclude that depression occurs later and as a consequence of sex and drug abuse because, in their study, depression itself did not increase the risk of sex and drug abuse; instead, those with riskier behaviors had a stronger association with depression [17]. In our study, while we could not deduce causality in the relationship between sexual intercourse and depression in adolescents, it is possible that depression occurred after sexual intercourse because the survey asked about depression only within the past year while there was no such limitation on their reports of sexual intercourse, although the cross-sectional study has limitation on making clear the order of incidence.

As the KYRBS did not use an instrument to measure depression severity, we divided the subjects into two groups based on suicidality: depression without suicidality and depression with suicidal ideation, planning, and attempts. The higher ORs of depression with suicidality indicates a possible positive correlation, in that higher amounts of sexual behavior could increase the severity of depression. Thus, appropriately managing adolescents with sexual experience is important to prevent them from experiencing severe depression. Previous studies have suggested the need for abstinence programs for students; these include encouraging the delay of sexual intercourse in early adolescents on the grounds that it is associated with depression [18]. Conversely, some researchers are of the opinion that only teaching about abstinence is not effective and that there is a need for more comprehensive research to plan proper sexual education that prevents depression in adolescents [19]. In the present study’s subgroup analysis, there was no difference in depression prevalence between subjects in early and late adolescence, which also supports the idea that just delaying sexual experience is not the solution. Further, in the subgroup analysis with sex education, the prevalence of depression was higher in the group that had received sex education than the group that had not. Thus, conventional sex education in South Korea might not be successful at preventing depression associated with sexual intercourse. Previous research on Korean sex education suggest that two-thirds of sex education was done by uneducated teachers [20]. Additionally, the main educational contents remain superficial and comprise formal theories that can easily induce negative perspectives on adolescent sexual behaviors. Simply suggesting delaying sexual intercourse or portraying it negatively not only fails to prevent depression but may even induce depressive symptoms. Therefore, there is a need for appropriate sex education, which includes helpful information, such as that sexual behavior is a natural part and a fundamental aspect of life, and teaches students how to respond to emotional changes that occur after sexual activities [21]. Proper sex education should also contain suggestions on how to manage the depression that could occur after sexual behavior, as our results indicate that the correlation between depression and sexual behavior may lead to further impulsive and unsafe sexual behavior [22].

This study has several limitations. First, the use of cross-sectional data prevents causal inferences regarding the relationship between sexual intercourse and depression prevalence in adolescents. Second, as the data were collected through a self-reported web-based survey, the possibility of recall bias or the subjects having misunderstood some questions cannot be eliminated. In particular, the answers about socioeconomic status might not be accurate because the subjects may not fully understand their parents’ economic condition. Conversely, the results are based on subjectively reported economic status, which can more accurately reflect depressive feelings than objectively reported data. Third, as the survey targeted adolescents, the questionnaire was kept deliberately simple and did not include a scale for assessing depressive symptoms.

## 5. Conclusions

In conclusion, this study identified the relationship between sexual behavior and the prevalence of depression; adolescents with experience of sexual intercourse were more likely to have depression with suicidality. There is a need for further research using prospective designs to identify cause and effect in this relationship. Further, the formulation of appropriate sex education policies based on a comprehensive understanding of the relationship between the variables is necessary to alleviate the effect of sexual intercourse on depression in adolescents.

## Figures and Tables

**Figure 1 ijerph-18-04228-f001:**
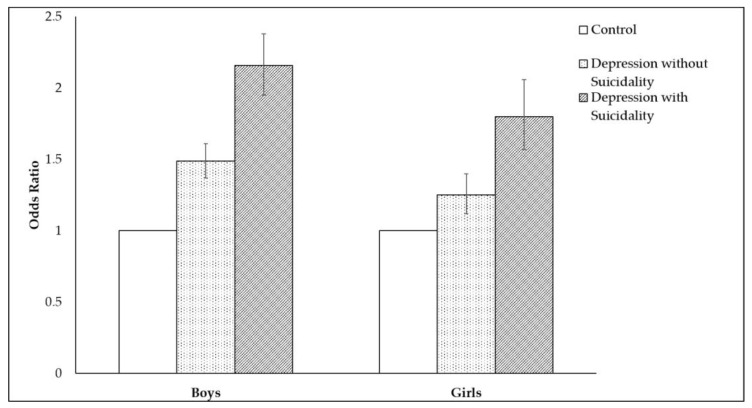
Odds ratios of depression without suicidality and depression with suicidality. The group without depression serves as the reference.

**Table 1 ijerph-18-04228-t001:** Socioeconomic and health-related characteristics according to the presence/absence of depression.

Variables	Boys (*n* = 91,309)					Girls (*n* = 87,355)				
	Depression		No Depression		*p*-Value	Depression		No Depression		*p*-Value
	*n*	(%)	*n*	(%)		*n*	(%)	*n*	(%)	
Sexual Intercourse Experience					<0.0001					<0.0001
No	16,607	(19.5)	68,468	(80.5)		27,063	(32.0)	57,470	(68.0)	
Yes	2252	(36.1)	3982	(63.9)		1510	(53.5)	1312	(46.5)	
Age					<0.0001					<0.0001
12–15	10,116	(19.0)	43,194	(81.0)		16,126	(31.5)	35,017	(68.5)	
16–18	8743	(23.0)	29,256	(77.0)		12,447	(34.4)	23,765	(65.6)	
Parents					<0.0001					<0.0001
Both	15,064	(20.1)	59,742	(79.9)		24,400	(32.2)	51,434	(67.8)	
Single	969	(24.4)	3001	(75.6)		1456	(38.1)	2365	(61.9)	
Neither	2826	(22.5)	9707	(77.5)		2717	(35.3)	4983	(64.7)	
Economic Status					<0.0001					<0.0001
Low	3423	(29.1)	8351	(70.9)		5329	(43.6)	6889	(56.4)	
Middle	7754	(19.2)	32,716	(80.8)		13,355	(30.9)	29,799	(69.1)	
High	7682	(19.7)	31,383	(80.3)		9889	(30.9)	22,095	(69.1)	
Grade					<0.0001					<0.0001
High	6795	(18.4)	30,044	(81.6)		9238	(28.4)	23,274	(71.6)	
Middle	4978	(19.3)	20,796	(80.7)		8275	(31.0)	18,383	(69.0)	
Low	7086	(24.7)	21,610	(75.3)		11,060	(39.2)	17,125	(60.8)	
Alcohol Consumption					<0.0001					<0.0001
Never	8293	(16.0)	43,477	(84.0)		15,277	(27.2)	40,845	(72.8)	
Yes	10,566	(26.7)	28,973	(73.3)		13,296	(42.6)	17,937	(57.4)	
Smoking Experience					<0.0001					<0.0001
No	13,617	(18.3)	60,710	(81.7)		25,159	(31.1)	55,830	(68.9)	
Yes	5242	(30.9)	11,740	(69.1)		3414	(53.6)	2952	(46.4)	
Physical Activity					<0.0001					<0.0001
Low	9196	(20.0)	36,811	(80.0)		21,619	(32.2)	45,589	(67.8)	
High	9663	(21.3)	35,639	(78.7)		6954	(34.5)	13,193	(65.5)	
Perceived Stress Level					<0.0001					<0.0001
Low	1376	(5.9)	21,977	(94.1)		871	(7.7)	10,477	(92.3)	
Middle	5919	(14.9)	33,675	(85.1)		6844	(19.9)	27,608	(80.1)	
High	11,564	(40.8)	16,798	(59.2)		20,858	(50.2)	20,697	(49.8)	
Self-Reported Health Status					<0.0001					<0.0001
High	12,567	(17.9)	57,755	(82.1)		15,307	(26.6)	42,194	(73.4)	
Middle	4429	(27.3)	11,799	(72.7)		9166	(40.5)	13,458	(59.5)	
Low	1863	(39.1)	2896	(60.9)		4100	(56.7)	3130	(43.3)	
Sex Education					0.2458					0.3517
Yes	14,310	(20.6)	55,267	(79.4)		22,897	(32.6)	47,262	(67.4)	
No	4549	(20.9)	17,183	(79.1)		5676	(33.0)	11,520	(67.0)	
Participants	18,859	(20.7)	72,450	(79.3)		28,573	(31.3)	58,782	(64.4)	

Variables are presented as numbers and percentages.

**Table 2 ijerph-18-04228-t002:** Results of the multivariate logistic regression analysis of the association between sexual behavior and depression.

Variables	Boys		Girls	
	Depression		Depression	
	AdjustedOR	95%CI	AdjustedOR	95%CI
Sexual Intercourse Experience				
No	1.00		1.00	
Yes	1.71	(1.59–1.83)	1.47	(1.33–1.61)
Age				
12–15	1.00		1.00	
16–18	0.95	(0.91–0.99)	0.83	(0.79–0.86)
Parents				
Both	1.00		1.00	
Single	1.05	(0.96–1.15)	1.03	(0.94–1.13)
Neither	1.13	(1.07–1.20)	1.09	(1.02–1.16)
Economic Status				
Low	1.00		1.00	
Middle	0.79	(0.74–0.83)	0.80	(0.76–0.84)
High	0.90	(0.85–0.96)	0.91	(0.86–0.96)
Grade				
High	1.00		1.00	
Middle	1.07	(1.02–1.13)	1.13	(1.08–1.18)
Low	1.18	(1.13–1.24)	1.32	(1.27–1.38)
Alcohol Consumption				
Never	1.00		1.00	
Yes	1.47	(1.41–1.53)	1.60	(1.54–1.66)
Smoking Experience				
No	1.00		1.00	
Yes	1.34	(1.27–1.41)	1.51	(1.41–1.62)
Physical Activity				
Low	1.00		1.00	
High	1.25	(1.20–1.31)	1.23	(1.18–1.28)
Perceived Stress Level				
Low	1.00		1.00	
Middle	0.28	(0.27–0.29)	0.28	(0.27–0.29)
High	0.10	(0.10–0.11)	0.10	(0.09–0.11)
Self-Reported Health Status				
High	1.00		1.00	
Middle	1.27	(1.21–1.33)	1.39	(1.34–1.45)
Low	1.77	(1.64–1.90)	2.11	(1.99–2.24)
Sex Education				
Yes	1.00		1.00	
No	0.93	(0.88–0.97)	0.92	(0.88–0.97)

OR: Odds ratio, CI: Confidence interval.

**Table 3 ijerph-18-04228-t003:** Subgroup analysis of the association between sexual behavior and depression stratified by sociodemographic variables.

Variables	Boys			Girls		
	Sexual Intercourse (−)	Sexual Intercourse (+)		Sexual Intercourse (−)	Sexual Intercourse (+)	
	AdjustedOR	AdjustedOR	95%CI	AdjustedOR	AdjustedOR	95%CI
Age						
12–15	1.00	1.79	(1.58–2.01)	1.00	1.46	(1.22–1.75)
16–18	1.00	1.69	(1.55–1.84)	1.00	1.51	(1.34–1.69)
Parents						
Both	1.00	1.70	(1.57–1.84)	1.00	1.49	(1.34–1.65)
Single	1.00	2.40	(1.81–3.19)	1.00	1.69	(1.19–2.40)
Neither	1.00	1.56	(1.33–1.84)	1.00	1.23	(0.93–1.62)
Economic Status						
Low	1.00	1.93	(1.64–2.26)	1.00	1.45	(1.20–1.74)
Middle	1.00	1.73	(1.54–1.94)	1.00	1.57	(1.35–1.82)
High	1.00	1.60	(1.45–1.78)	1.00	1.36	(1.14–1.63)
Grade						
High	1.00	1.81	(1.61–2.04)	1.00	1.42	(1.18–1.70)
Middle	1.00	1.57	(1.36–1.80)	1.00	1.65	(1.36–2.00)
Low	1.00	1.71	(1.52–1.91)	1.00	1.40	(1.22–1.61)
Alcohol Consumption						
Never	1.00	1.75	(1.51–2.03)	1.00	1.65	(1.33–2.06)
Yes	1.00	1.72	(1.59–1.85)	1.00	1.44	(1.29–1.60)
Smoking Experience						
Never	1.00	1.90	(1.72–2.10)	1.00	1.56	(1.37–1.78)
Yes	1.00	1.59	(1.45–1.75)	1.00	1.39	(1.20–1.60)
Physical Activity						
Low	1.00	1.82	(1.64–2.01)	1.00	1.38	(1.24–1.54)
High	1.00	1.62	(1.47–1.78)	1.00	1.76	(1.44–2.15)
Perceived Stress Level						
Low	1.00	1.59	(1.44–1.75)	1.00	1.39	(1.22–1.57)
Middle	1.00	1.69	(1.51–1.89)	1.00	1.51	(1.28–1.79)
High	1.00	2.31	(1.91–2.80)	1.00	1.76	(1.19–2.58)
Self-Reported Health Status						
High	1.00	1.66	(1.52–1.80)	1.00	1.49	(1.31–1.70)
Middle	1.00	1.90	(1.63–2.23)	1.00	1.52	(1.28–1.81)
Low	1.00	1.72	(1.35–2.20)	1.00	1.27	(0.98–1.63)
Sex Education						
Yes	1.00	1.79	(1.65–1.94)	1.00	1.58	(1.41–1.77)
No	1.00	1.52	(1.33–1.73)	1.00	1.16	(0.95–1.43)

OR: Odds ratio, CI: Confidence interval.

**Table 4 ijerph-18-04228-t004:** Subgroup analysis of the association between depression with/without suicidality and sexual behavior.

Variables	Boys				Girls			
	Depression without Suicidality		Depression with Suicidality		Depression without Suicidality		Depression with Suicidality	
	(*n* = 6290)		(*n* = 12,569)		(*n* = 11,290)		(*n* = 17,283)	
	Adjusted OR	95%CI	Adjusted OR	95% CI	AdjustedOR	95% CI	Adjusted OR	95% CI
SexualIntercourse Experience								
No	1.00		1.00		1.00		1.00	
Yes	1.49	(1.37–1.61)	2.16	(1.95–2.38)	1.25	(1.12–1.40)	1.80	(1.57–2.06)

OR: Odds ratio, CI: Confidence interval.

## Data Availability

The data analyzed in this study were taken from 2017-2019 KYRBS which is available to the public with academic purpose. The data can be downloaded in the official website (https:// http://www.kdca.go.kr/yhs/, accessed on 15 January 2021).

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
