# Peer review of "Association between Sexual Behavior and Depression in South Korean Adolescents: A Cross-Sectional Study"

_ijerph, 2021, doi:10.3390/ijerph18084228_

Round 1
Reviewer 1 Report
Line numbering is generally is generally presented on the left side.
Table one should be broken down into multiple tables (at least 3).
All tables and figures should have an introduction followed by a summary instead of summarizing before the table.
The references should follow a standard format for the style chosen. For example, articles in journals should be in lower case.
For citations, the first should list all authors before using et al.
The word percent should be written when in text, and use the symbol in tables and figures.
Author Response
Thank you for your helpful and kind comments.
Our manuscript had formatting problem, as you mentioned, thus we have revised the format of whole manuscript and the reference.
Also, we got the English editing by Editage.
Please see the attachment

Reviewer 2 Report
This is a very important study and I believe has the potential of making a strong contribution to the extant literature. There are a few major problems that have to be addressed but they are based in the analysis and not the actual study and can be addressed. I strongly encourage the authors to make the needed changes so that this paper can go forward. I have left almost 30 comments in the paper to provide suggestions for improvement.

Author Response
Thank you for your kind and detailed comments.
We have revised the manuscript and answered your questions.
We also got English editing by Editage.
Please see the attatchment.

Round 2
Reviewer 2 Report
Well done. I believe it is will make a meaningful contribution.